# Effects of Elicitation on *Abeliophyllum distichum* Leaf Callus and Changes in Verbascoside Content

**DOI:** 10.3390/plants14091386

**Published:** 2025-05-04

**Authors:** Daeho Choi, Yong-Woo Park, Jungmok Kang, Eun-Suk Jung, Hwayong Lee

**Affiliations:** 1Forest Bio Center, Chungcheongbuk-do Forest Environment Research Center, Okcheon 29061, Republic of Korea; eoghchoi@korea.kr (D.C.); pywcan@korea.kr (Y.-W.P.); kjm1111@korea.kr (J.K.); winjung7@korea.kr (E.-S.J.); 2Department of Forest Science, Chungbuk National University, Cheongju 28644, Republic of Korea

**Keywords:** callus, elicitor, methyl jasmonate, salicylic acid

## Abstract

*Abeliophyllum distichum* is a monotypic species in the family Oleaceae that contains a range of phenolic compounds and components such as coumaric acid, catechin, and verbascoside, the latter of which is a major candidate of commercial interest. In this study, we assessed the potential for producing verbascoside using callus culture. To enhance callus productivity in this regard, we evaluated the efficacy of treatment with the elicitors salicylic acid (SA) and methyl jasmonate (MeJA) based on changes in verbascoside content with callus development using Petri dish cultures. Whereas the initial content of verbascoside in *A. distichum* callus was approximately 50 mg/g, in response to treatment with 50 μM MeJA, we detected an increase to approximately 97.05 mg/g. In contrast, treatment with SA had no significant effects on verbascoside content. In addition, we found that the fresh weight of callus receiving elicitor treatment was lower than that of control callus. Conversely, however, in bioreactor cultures, the fresh weight of callus following treatment with 50μM MeJA for 1 week was higher than that of control callus, and the content of verbascoside in callus treated with 50 μM MeJA was higher than that in control callus. Our findings in this study thus indicate that with appropriate elicitation, the production of verbascoside by *A. distichum* callus pieces can be enhanced.

## 1. Introduction

*Abeliophyllum distichum* Nakai, a deciduous shrub in the family Oleaceae, is native to Korea, in which it is distributed primarily in the central regions [1]. *A. distichum* has received considerable attention with respect to its potential use in medicinal materials, cosmetics, and functional foods, with reported anti-wrinkle, anti-inflammatory, anti-obesity, and skin-whitening properties [2,3,4]. Its reported active compounds include verbascoside, rutin, catechin, chlorogenic acid, and coumaric acid [5,6,7].

Among the numerous compounds such as coumaric acid, catechin identified in *A. distichum*, verbascoside, a caffeoyl phenylethanoid glycoside, also referred to as acteoside, is considered the key commercially important compound [8]. Although found primarily in species of the genus *Verbascum* [9], verbascoside has also been identified in over 200 plant species among 23 genera [10]. Moreover, in addition to its aforementioned activities, verbascoside has also been established to have neuroprotective effects [11,12], anti-inflammatory properties [13,14], antibacterial activity [15,16], and antiviral properties [17], and is currently being developed as a pharmaceutical material, owing to its wide-ranging pharmacological effects and minimal side effects in humans [18]. The leaves of *A. distichum* contain approximately 162.11 mg/g of verbascoside when extracted using ethanol and under reflux [6]. This content is higher than that of *Olea europaea* leaves, which contain approximately 43.6 mg/kg of verbascoside, and the flowers of *Buddleja officinalis*, which contain 20–35 mg/g [19,20]. Therefore, verbascoside is emerging as a potential marker compound for the industrialization of *A. distichum*.

Methods used for obtaining plant-derived compounds include callus, plant cell, and tissue suspension cultures, which can address the issue of limited sources of plant material and enable efficient and cost-effective mass production [21]. However, the cost associated with setting up cultivation facilities such as bioreactors is a limitation that needs to be overcome [22]. Approaches based on plant tissue culture can facilitate the extraction of larger amounts of higher quality secondary compounds [23]. Techniques such as callus and in vitro propagation can ensure high rates of secondary compound production without the influence of the atmosphere [24,25], the former of which refers to the proliferation of cells in wounded explant tissues separated from plant organs [26]. Callus culture, which can be optimized by controlling nutrients, plant growth regulators, and inducers, has an advantage, in that it can facilitate the efficient harvest of biomass within relatively short periods of time (approx. 4 to 8 weeks) via cell suspension culture [27].

The production of biosynthetic compounds in callus cultures is dependent on both callus growth and the content of secondary metabolites [28]. Accordingly, to meet the high demand for secondary metabolites, content enhancement studies are necessary. In this regard, elicitation is a well-established technique for biochemical manipulation and metabolic pathway enhancement via stress induction in plants [29]. Elicitor treatment can contribute to promoting the production of secondary metabolites in plants and callus pieces, with compounds such as salicylic acid (SA), methyl jasmonate (MeJA), chitosan, and cerebroside being commonly used in this regard [30]. SA and MeJA are recentely studied elicitors for promoting the production of secondary metabolites in callus cultures [31,32]. However, while the production of verbascoside via callus cultivation has been studied in a range of plants, including *Verbena officinalis*, *Verbascum thapsus*, and *V. scamandri* [33,34,35], further studies on its production in *A*. *distichum* is still lacking.

In this study, we thus sought to establish an efficient callus-based production system by assessing the efficacy of different types and concentrations of elicitors in enhancing the growth of *A. distichum* callus pieces, with a particular focus on the content of the target compound verbascoside using Petri dish cultures. Additionally, using bioreactor cultures, we compared the effects of the duration of elicitor treatment to determine the optimal conditions for efficient production of this secondary compound.

## 2. Results

### 2.1. Salicylic Acid and Methyl Jasmonate Elicitation in Petri Dishes

#### 2.1.1. Fresh and Dry Weights of Callus

To determine the optimal elicitor, we cultured callus pieces on media treated with either SA or MeJA for 4 weeks. Notably, whereas the control callus pieces were characterized by robust growth without any signs of browning, the SA- and MeJA-treated callus exhibited browning accompanied by tissue necrosis, the intensity of which was more pronounced in callus treated with higher concentrations of these elicitors (Figure 1).

After 4 weeks of cultivation, the control callus was established to have the highest fresh and dry weights of 1.22 ± 0.37 and 0.08 ± 0.02 g, respectively (Figure 2). Comparatively, the fresh and dry weights of SA- and MeJA-elicited callus were significantly lower, with fresh and dry weights of 0.59 ± 0.15 and 0.05 ± 0.01 g and 0.54 ± 0.11 and 0.04 ± 0.01 g for callus elicited with 50 and 100 μM SA, respectively, and fresh and dry weights of 0.47 ± 0.18 and 0.04 ± 0.02 g and 0.35 ± 0.08 and 0.03 ± 0.01 g for callus elicited with 50 and 100 μM MeJA, respectively.

#### 2.1.2. Verbascoside Content

We detected a statistically significant difference among the assessed treatments with respect to the verbascoside contents of callus (Figure 3). After 4 weeks of cultivation, the highest verbascoside content of 94.39 ± 36.82 mg/g of dry callus was recorded in the 50 μM MeJA-elicited callus (Appendix A). Moreover, the levels in callus were consistently the highest throughout the period of cultivation. Whereas lower levels were obtained in the 100 μM MeJA-elicited callus, these were still higher than those detected in callus treated with 50 and 100 μM SA, which were even lower than the contents of verbascoside in the control callus.

### 2.2. Methyl Jasmonate Elicitation in Bioreactors

#### 2.2.1. Fresh and Dry Weight of Callus

To assess productivity based on the elicitor treatment duration, callus pieces were cultured in bioreactors for 4 weeks. We accordingly found that with an extension of the duration of MeJA treatment, there was an increasing in browning callus tissue and growth was progressively inhibited, with an eventual cessation of new callus formation (Figure 4).

The highest fresh weight of the callus (approx. 39.43 ± 9.00 g) was obtained following treatment with MeJA for 1 week, whereas weights of approximately 23.07 ± 5.00, 22.24 ± 2.25, and 23.86 ± 3.07 g were recorded following treatments for 2, 3, and 4 weeks, respectively, which were all lower than the 35.32 ± 9.00 g of control callus, although the differences were not statistically significant (Figure 5).

The dry weights of the callus followed a trend similar to that of the fresh weight, being highest for callus treated with MeJA for 1 week, at approximately 3.20 ± 0.89 g, compared with 1.68 ± 0.52, 1.59 ± 0.58, and 1.28 ± 0.13 g obtained for callus treated with MeJA for 2, 3, and 4 weeks, respectively, which were again lower than the 2.35 ± 0.55 g obtained for the control callus (Figure 5).

#### 2.2.2. Verbascoside Content

For the same dry weight of callus, we recorded verbascoside contents of approximately 131.55 ± 17.52, 87.36 ± 9.98, 135.03 ± 12.14, and 112.43 ± 25.50 mg/g of dry callus, in callus treated with MeJA for 1, 2, 3, and 4 weeks, respectively, all of which higher compared with the value recorded in the control callus at approximately 62.87 ± 22.51 mg/g of dry callus (Figure 6 and Appendix A).

## 3. Discussion

In response to elicitor treatment using 50 μM MeJA, we obtained a verbascoside content of approximately 55 mg/ g of dry callus in *A. distichum* callus cultured in Petri dishes, which increased to approximately 97.05 mg/g of dry callus when cultured in Petri dishes for 3 weeks. It has been established that quantities of secondary metabolites in callus can differ depending on the induced tissues, medium composition, and cultural environment [36]. Comparatively, the verbascoside content obtained in this study was higher than the levels reported previously (7.15 mg/g of fresh callus). In the previous study, the extraction was performed using non-dried callus, which had a high moisture content, leading to a significant discrepancy between the measured and actual verbascoside content [7]. This suggests the potential feasibility of establishing a mass production system for verbascoside using callus derived from *A. distichum*. Callus growth can be represented by a sigmoid standard curve divided into six phases, namely, the lag, exponential, linear, deceleration, stationary, and decline phases [37]. However, in the present study, we failed to observe a deceleration phase during the 4-week culture period in bioreactor, thus indicating the potential to enhance the rate of callus growth, and thereby promote an increase the production of verbascoside.

Our assessment of the efficacy of two different elicitors revealed that whereas SA treatment had no significant effects with respect to promoting the verbascoside content in callus, MeJA induced increases in the contents of this secondary metabolite, although tended to inhibit callus development, thereby resulting in a lower biomass. This is consistent with the findings of Danaee et al. [38], who showed that MeJA treatment inhibited callus growth, whilst promoting increases in the contents secondary metabolites. Given the high levels of verbascoside detected in MeJA-treated callus, we speculate that it would be beneficial to initially culture *A. distichum* callus in a medium lacking MeJA for a certain period of time to facilitate the accumulation of biomass and subsequently initiate elicitor treatment with MeJA to enhance the verbascoside content. In the 2 weeks MeJA treatment group, the increase in verbascoside content was relatively low. To elucidate the underlying cause, further experiments involving metabolic pathway analysis and identification of genes encoding related enzymes are warranted. In other studies, it has been found that treatment of *Buddleja cordata* callus culture with 50 μM MeJA resulted in a 2.13-fold increase in verbascoside content [39], whereas in a *Rehmannia glutinosa* adventitious root culture, SA treatment showed no significant effects in enhancing the contents of this metabolite, although treatment with 200 μM MeJA resulted in more than 10-fold increase in verbascoside [30]. Similarly, we found that whereas SA treatment was ineffective in promoting an increase in verbascoside content, MeJA was effective in this regard.

In previous studies, it has generally been reported that treatment with MeJA has the effect of inhibiting callus growth, resulting in reductions in fresh and dry weights [38,40]. Conversely, however, Chavan et al. [41] have reported that low concentrations of MeJA (approx. 100 μM) promoted an increase in the fresh weight of *Salacia chinensis* callus culture, whereas when applied at higher concentrations (e.g., 250 μM), this hormone reduced callus growth. In the bioreactor cultivation assessed in the present study, compared with the control treatment, we obtained higher callus fresh weights in response to the treatment with MeJA for 1 week, whereas lower weights were obtained when extending the period of cultivation to 2, 3, and 4 weeks, thus providing evidence that a prolongation of MeJA treatment has the effect of inhibiting callus growth. Nevertheless, despite this growth inhibition, callus cultivated in the presence of MeJA for the assessed time periods were found to have verbascoside contents higher than those detected in the control callus, thus indicating that culturing *A. distichum* callus for 3 weeks in medium lacking elicitors prior to the addition of 50 μM MeJA for 1 week can potentially increase verbascoside yield. The contents of metabolites synthesized during cell culture can differ depending on the duration and concentration of the elicitor treatment [42]. Accordingly, given that shorter-term elicitor treatment has been shown to enhance verbascoside content, as observed in plants such as *B. cordata* [39] and *R. glutinosa* [43], further research on similar short-term treatments is warranted.

## 4. Materials and Methods

### 4.1. Plant Materials and Culture Conditions

Leaves from the shoots of 3-year-old plants of the *A. distichum* cultivar ‘Kkoribyul’ were collected from an experimental field of the Chungbuk Forest Biocenter, Chungcheongbuk Province, Korea. Initially, the leaves were rinsed with sterilized distilled water for 15 min and 70% ethanol for 15 min. After rinsing, the explants were immersed in 2% (*v*/*v*) sodium hypochlorite solution for 15 min and thereafter rinsed three times with sterile water. Having trimmed the explants to a length of approximately 0.5 cm they were subsequently transferred to MS medium [44] supplemented with 1.0 mg/L 2,4-dichlorophenoxyacetic acid (2,4-D, MB Cell, Seoul, Republic of Korea), 30 g/L sucrose (MB Cell, Seoul, Republic of Korea), and 2.3 g/L gelrite (MB Cell, Seoul Republic of Korea) to induce callus formation. Prior to autoclaving at 121 °C for 15 min, the pH of the medium was adjusted to 5.8 using 1 N NaOH. In addition, to regulate callus browning, 10.0 mg/L thiamine-HCl (Duchefa Biochemie, Haarlem, Netherland) was added to the sterilized medium. Callus regenerated in vitro was sub-cultured in fresh medium at 4-week intervals. During cultivation, the cultures were maintained in the dark at a temperature of 25 ± 1 °C.

### 4.2. Elicitation with Salicylic Acid and Methyl Jasmonate

Samples (0.3 g) of callus were cultured in Petri dishes on MS medium supplemented with 1.0 mg/L 2,4-D, 30 g/L sucrose, 2.3 g/L gelrite, and 10.0 mg/L thiamine-HCl in combination with 50 or 100 μM SA (Sigma-Aldrich, St. Louis, MO, USA) or MeJA (Bedoukian Research, Danbury, CT, USA) for 4 weeks. with each treatment being replicated 10 times. Over the 4-week period of culture, callus from each treatment was collected at weekly intervals, the fresh weight of which was measured, with the dry weight of callus being subsequently determined after 48 h of drying at 40 °C. Both control and elicited callus were assessed for the production of verbascoside.

### 4.3. Methyl Jasmonate Elicitation Period in Bioreactor

Callus pieces (10 g/L) were cultured in 2L MS medium supplemented with 1.0 mg/L 2,4-D and 30 g/L sucrose for 4 weeks in a 5L balloon-type air-lift bioreactor (Dongmyung Chemical, Seoul, Republic of Korea). Prior to autoclaving at 121 °C for 15 min, the pH of the medium was adjusted to 5.8 using 1 N NaOH. To regulate callus browning, 10.0 mg/L thiamine-HCl was added to the sterilized medium. To determine the optimal elicitation period, 50 μM MeJA treatments were applied for 1, 2, 3, or 4 weeks during the cultivation period, with each treatment being replicated three times. Cultivation in bioreactors was conducted in the dark at 25 ± 1 °C, during which, sterilized air was supplied at 0.1 vvm (air volume/medium volume/min). After 4 weeks, callus from each treatment was collected, the fresh weights were measured, dry callus weights were determined after drying at 40 °C for 48 h, and we assessed the contents of verbascoside.

### 4.4. Analysis of Verbascoside Content

#### 4.4.1. Extraction of Verbascoside from Callus

The dried samples from each treatment were subjected to ultrasonic extraction for 90 min at 40 °C using 70% ethanol at a sample to ethanol ratio of 1:100. The extract thus obtained was then centrifuged at 4000 rpm for 20 min (Centrifuge 5920R, Eppendorf, Hamburg, Germany) and used for subsequent analysis.

#### 4.4.2. Determination of Verbascoside Content Using Reversed-Phase High-Performance Liquid Chromatography (RP-HPLC)

A standard preparation of verbascoside (1 mg) was weighed and dissolved in 1 mL of acetonitrile (HPLC grade) to yield a 1 mg/mL stock solution, which was subsequently serially diluted to produce standard solutions with concentrations of 125, 250, 500, and 1000 mg/L, which were used for further analyses.

The method we used to analyze the contents of verbascoside was based on a procedure outlined by Rahmat et al. [45], with slight modification. Prior to injection into an RP-HPLC system, the extract and standard preparations were filtered through a 0.45-μM membrane filter (PVDF: HENKE-JECT, Tuttlingen, Germany). This system consists of a reversed-phase Shim-pack VP-ODS C8 column (250 × 4.6 mm, particle size 5.0 microns) and a Thermo Ultimate 3000 HPLC system (Thermo Scientific, MA, USA), equipped with a solvent pump (LPS-3400SD), an autosampler (WPS-3000TSL), a column oven (TCC-3000SD), and a detector (VWD-3400RS). Injected samples (10 µL) were subjected to gradient elution with a mixture of 0.1% formic acid in water and acetonitrile, the ratio of these two solvents was applied as follows: 0 min, 90:10; 35 min, 50:50; 40 to 45 min, 5:95; 45 to 48 min, and 90:10, at a flow rate of 1.0 mL/min. Verbascoside concentrations were determined at 320 nm and the retention time of verbascoside was approximately 15.35 min. Chromatographic analysis was performed using Chromeleon software 7.3.2. (Thermo Scientific, Waltham, MA, USA).

### 4.5. Statistical Analysis

Data were analyzed using a one-way analysis of variance, with the means being compared using Duncan’s multiple range test at a significance level of 0.05, using R version 4.2.2.

## 5. Conclusions

This study demonstrates that methyl jasmonate (MeJA) is an effective elicitor for enhancing verbascoside production in *A. distichum* callus cultures. While salicylic acid (SA) had no significant effect, treatment with 50 μM MeJA significantly increased verbascoside content, despite reductions in biomass. In bioreactor cultures, applying MeJA for 1 or 3 weeks resulted in the highest verbascoside yields, indicating that a short-term elicitation strategy may optimize both biomass and metabolite production. Notably, this study also confirms that the verbascoside content in callus cultures is comparable to that reported in leaves, suggesting the potential feasibility of using callus as an alternative source for verbascoside production. These findings provide a basis for developing an efficient in vitro production system for verbascoside. However, this experiment was conducted in triplicates for each treatment, and it is believed that the variability was relatively high. Increasing the number of replicates and conducting additional experiments would likely lead to more accurate results. Further studies should focus on minimizing variability and validating the reproducibility of these findings under a broader range of conditions. Further studies will also be required to optimize detailed processes for potential commercial application, and it will be necessary to develop superior varieties through gene editing.

## Figures and Tables

**Figure 1 plants-14-01386-f001:**
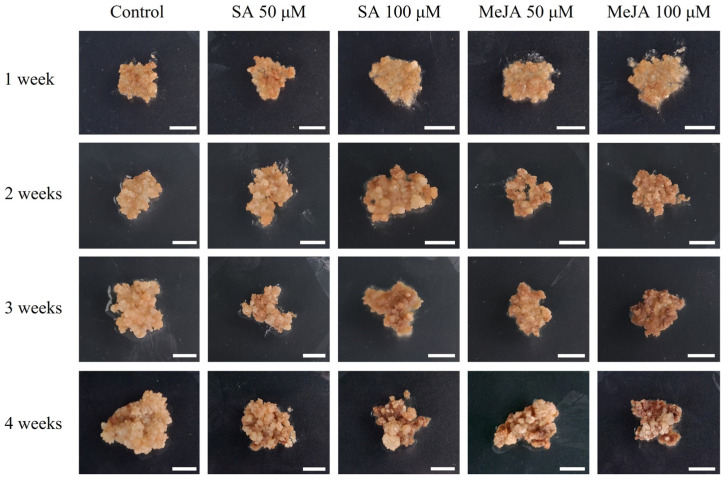
Callus of *Abeliophyllum distichum* by concentration of SA and MeJA treatment in Petri dishes. SA and MeJA were applied at concentrations of 50, 100 µM, and callus growth was monitored weekly over a 4-week cultivation period. Bars represent 1 cm.

**Figure 2 plants-14-01386-f002:**
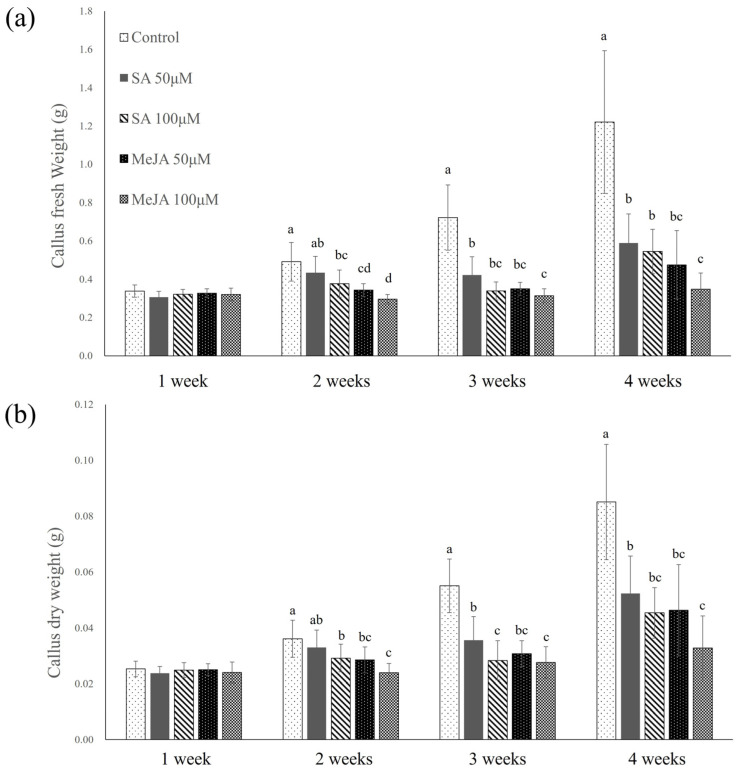
(**a**) Fresh weight and (**b**) dry weight of *Abeliophyllum distichum* callus by concentration of SA and MeJA treatment in Petri dishes. SA and MeJA were applied at concentrations of 50, 100 µM, and callus growth was monitored weekly over a 4-week cultivation period. Different letters above the columns indicate a significant difference (*p* < 0.05), and error bars indicate ± SD.

**Figure 3 plants-14-01386-f003:**
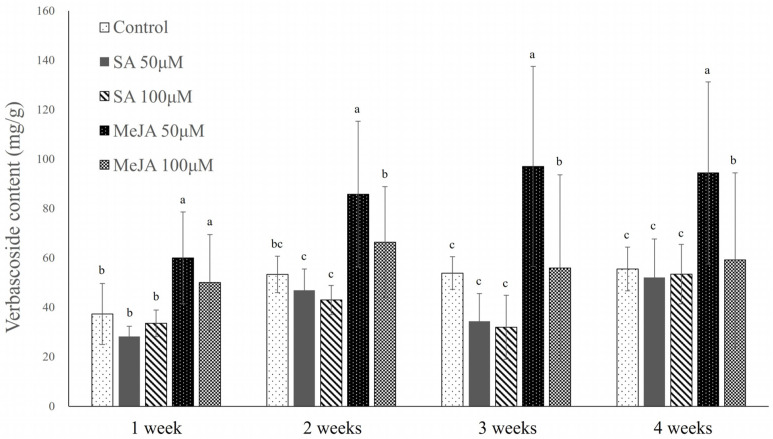
Verbascoside content of *Abeliophyllum distichum* callus by concentration of SA and MeJA treatment in Petri dishes. SA and MeJA were applied at concentrations of 50, 100 µM, and callus growth was monitored weekly over a 4-week cultivation period. Different letters above the columns indicate a significant difference (*p* < 0.05), and error bars indicate ± SD.

**Figure 4 plants-14-01386-f004:**
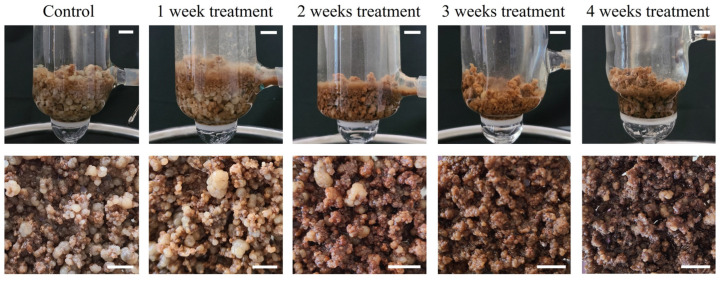
Callus of *Abeliophyllum distichum* by period of 50μM MeJA treatment in bioreactor. MeJA was applied for 1, 2, 3, or 4 weeks, and the measurements were taken after 4 weeks of cultivation. Bars represent 1 cm.

**Figure 5 plants-14-01386-f005:**
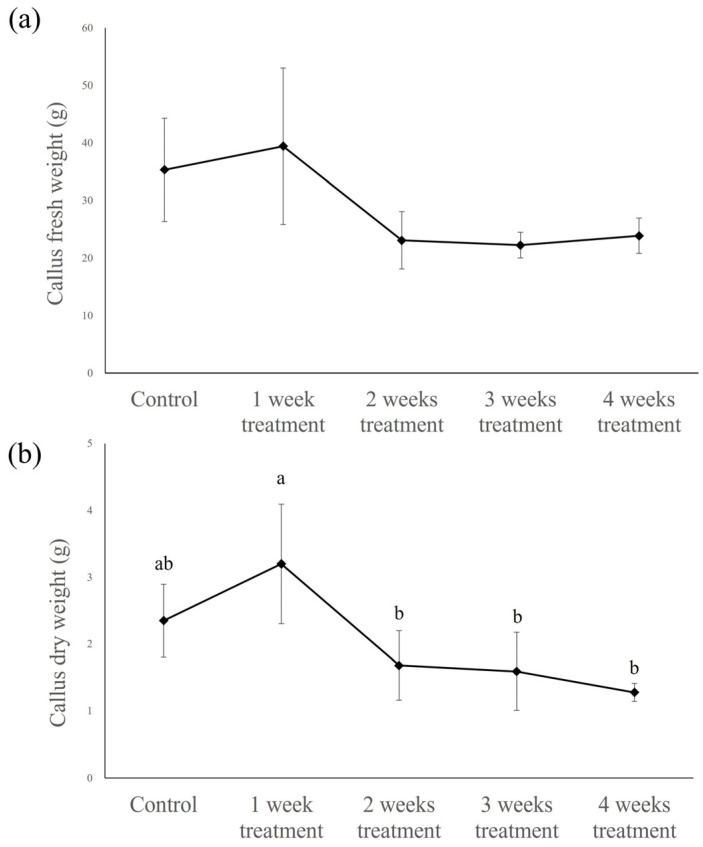
Fresh weight (**a**) and dry weight (**b**) of *Abeliophyllum distichum* callus by period of 50μM MeJA treatment in bioreactor. MeJA was applied for 1, 2, 3, or 4 weeks, and the measurements were taken after 4 weeks of cultivation. Different letters above the columns indicate a significant difference (*p* < 0.05), and error bars indicate ± SD.

**Figure 6 plants-14-01386-f006:**
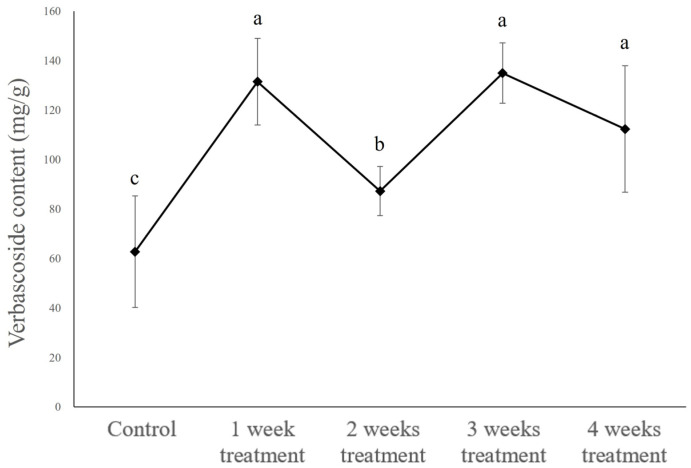
Verbascoside content of *Abeliophyllum distichum* callus by period of 50μM MeJA treatment in bioreactor. MeJA was applied for 1, 2, 3, or 4 weeks, and the measurements were taken after 4 weeks of cultivation. Different letters above the columns indicate a significant difference (*p* < 0.05), and error bars indicate ± SD.

## Data Availability

The data are contained within the article.

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
