# Peer review of "Effects of Elicitation on Abeliophyllum distichum Leaf Callus and Changes in Verbascoside Content"

_plants, 2025, doi:10.3390/plants14091386_

Round 1
Reviewer 1 Report
Comments and Suggestions for Authors
Dear authors of “Effects of Elicitation on Abeliophyllum distichum Leaf Calli and Changes in Verbascoside Contents”.
In general, your paper was interesting but not exciting. You did not convince me that verbascoside is important metabolite for production. Also, I did not find that production of this compound from callus is more efficient than from in vivo plants or roots or from in vitro propagated plants or transformed roots (rhizogenes). You also did not do and did not mention the possibility to enhance production of this metabolite by editing. It would be interesting to know about metabolic pathway of it and genes encoding it.
Also, you had a lot of typos and mistakes.
Already in Abstract on line #9 you wrote “Abelophyllum” in other cases it is Abeliophylum. On line #15 you wrote 50mM, which is not correct. And then on line #19 it is 50 mM. The same mistakes could be found on Fig.1, line #85, Fig. 3, 4, line #109, #122, line #127, 129, etc.
On line #78 you wrote “petri dishes”. It should be “Petri dishes”.
On line #199 “in in Petri dishes’
On line # 127, 129, etc. you are writing “25.5 mg/dry callus g”. It would be better 25.5 mg/g of dry callus. It is written in many places.
“Calli”. There is no such word. And plural form is not available. It is jargon and not acceptable in scientific publication. If you wish to indicate plural form you can write: callus pieces, callus clumps, etc.
On line #153, #167 you wrote names (Chavan et al.) and did not indicate Ref. #. More correct will be written Rahman et al [39} as on line #229.
Line #188. How you can rinse for 15 min with 70% ethanol? It would be dead tissue.
It is necessary to write “in vitro” italics: In vitro. Line #196. The same for in vivo.
Line #121. You wrote that you centrifuged at 4000 rpm. Better to write in g. Or you need to indicate what centrifuge you used.
Line #192. You don’t have reference for MS medium.
In Material and Methods, it is necessary to include from which company you bought MeJA, SA and growth regulators.
In References it is important for each reference to indicate https://doi.org.
In Text there is no need to write Figure 1. It can be (Fig. 1).
Line #226. Instead of concentration in ppm write in mg/l.
Comments on the Quality of English Language
See comments above
Author Response
<Reviewer 1>
Dear authors of “Effects of Elicitation on Abeliophyllum distichum Leaf Calli and Changes in Verbascoside Contents”.
In general, your paper was interesting but not exciting. You did not convince me that verbascoside is important metabolite for production. Also, I did not find that production of this compound from callus is more efficient than from in vivo plants or roots or from in vitro propagated plants or transformed roots (rhizogenes). You also did not do and did not mention the possibility to enhance production of this metabolite by editing. It would be interesting to know about metabolic pathway of it and genes encoding it.
Responses the Reviewer comment: Thank you for your comments. We have written the representative plants in which the content of verbascoside was reported in the manuscript and its content in A. distichum (Line 41). Therefore, we wrote in the introduction that A. distichum is competitive in producing verbascoside. In addition, the sentence that the verbascoside content in callus was higher than that in leaves previously reported was added to the conclusion, describing it as efficient (Line 281). Finally, it was mentioned that it would be necessary to breed superior varieties through gene editing in the future.
Also, you had a lot of typos and mistakes.
Already in Abstract on line #9 you wrote “Abelophyllum” in other cases it is Abeliophylum. On line #15 you wrote 50mM, which is not correct. And then on line #19 it is 50 mM. The same mistakes could be found on Fig.1, line #85, Fig. 3, 4, line #109, #122, line #127, 129, etc.
Response the Reviewer comment: "Abelophyllum" was corrected to "Abeliophyllum" and "micro mol" was corrected to "μM". Thank you.
On line #78 you wrote “petri dishes”. It should be “Petri dishes”.
Response the Reviewer comment: We Changed "petri" to "Petri"
On line #199 “in in Petri dishes’
Response the Reviewer comment: Corrected the typo in "in"
On line # 127, 129, etc. you are writing “25.5 mg/dry callus g”. It would be better 25.5 mg/g of dry callus. It is written in many places.
Response the Reviewer comment: Thanks for your comment. Corrected to "g of dry callus" (Line 93, 127, 129, 138)
“Calli”. There is no such word. And plural form is not available. It is jargon and not acceptable in scientific publication. If you wish to indicate plural form you can write: callus pieces, callus clumps, etc.
Response the Reviewer comment: Thanks for your comment. We changed "Calli" to "callus" or "callus pieces"
On line #153, #167 you wrote names (Chavan et al.) and did not indicate Ref. #. More correct will be written Rahman et al [39} as on line #229.
Response the Reviewer comment: Thanks for your comment. We corrected references
Line #188. How you can rinse for 15 min with 70% ethanol? It would be dead tissue.
Response the Reviewer comment: Explained that washing with 70% EtOH for 15 minutes may cause some tissue death but ensures cleaner samples, reducing contamination during culture, and this method was applied in this study
It is necessary to write “in vitro” italics: In vitro. Line #196. The same for in vivo.
Response the Reviewer comment: We italicized "in vitro"
Line #121. You wrote that you centrifuged at 4000 rpm. Better to write in g. Or you need to indicate what centrifuge you used.
Response the Reviewer comment: We added equipment details: (Centrifuge 5920R, Eppendorf, Hamburg, Germany) (Line 121)
Line #192. You don’t have reference for MS medium.
In Material and Methods, it is necessary to include from which company you bought MeJA, SA and growth regulators.
Response the Reviewer comment: Added references for MS medium, MeJA, SA, and the suppliers of growth regulators (Line 192, 201)
In References it is important for each reference to indicate https://doi.org.
Response the Reviewer comment: References have been corrected, and DOI has been added.
In Text there is no need to write Figure 1. It can be (Fig. 1).
Response the Reviewer comment: We changed "Figure" to "Fig.“
Line #226. Instead of concentration in ppm write in mg/l.
Response the Reviewer comment: We corrected "ppm" to "mg/L“

Reviewer 2 Report
Comments and Suggestions for Authors
The article investigates the effects of elicitation on the production of verbascoside in Abeliophyllum distichum callus cultures. Researchers assessed the efficacy of using salicylic acid (SA) and methyl jasmonate (MeJA) as elicitors to enhance verbascoside content.
Key words – The keywords must not be the same as in the title of the work. Please change: Abeliophyllum distichum; verbascoside.
The article is well-structured with clearly defined sections. However, I have a few comments on individual paragraphs.
Introduction
The literature review is adequate, providing a comprehensive background on Abeliophyllum distichum and verbascoside. However, there could be a more detailed discussion on recent advancements in elicitation techniques and their applications in similar studies to highlight the novelty of the research.
Also please expand the literature review to include more recent studies (years 2024-2025).
Results
The legend of all Figures should be shelf explanatory. Please add important information’s in accordance with Figure.
Discussion
Please provide a deeper discussion on the limitations and potential improvements in the methods.
Conclusions
Conclusions in its current form rather resembles a summary of results. Please shorten the description of the results, limiting yourself to the most important conclusions of the paper. Please add information on further research perspectives.
Author Response
<Reviewer 2>
The article investigates the effects of elicitation on the production of verbascoside in Abeliophyllum distichum callus cultures. Researchers assessed the efficacy of using salicylic acid (SA) and methyl jasmonate (MeJA) as elicitors to enhance verbascoside content.
Key words – The keywords must not be the same as in the title of the work. Please change: Abeliophyllum distichum; verbascoside.
Response the Reviewer comment: Thanks for your comment. We removed “Abeliophyllum distichum”, and “verbascoside” from the keywords and added “Methyl jasmonate” and “Salicylic acid”.
The article is well-structured with clearly defined sections. However, I have a few comments on individual paragraphs.
Introduction
The literature review is adequate, providing a comprehensive background on Abeliophyllum distichum and verbascoside. However, there could be a more detailed discussion on recent advancements in elicitation techniques and their applications in similar studies to highlight the novelty of the research.
Also please expand the literature review to include more recent studies (years 2024-2025).
Response the Reviewer comment: Thanks for your valuable comment. We compared the verbascoside contents reported in various plants with those found in A. distichum for similar studies, highlighting the competitive potential of A. distichum in verbascoside production (line 41). In addition, we added the recent studies of Song et al. (2024) [31] and Cuaspud et al. (2025) [32] who used SA and MeJA to increase secondary metabolite production in callus cultures (line 66).
Results
The legend of all Figures should be shelf explanatory. Please add important information’s in accordance with Figure.
Response the Reviewer comment: Thanks for your comment. We have improved the visibility of the legends and figures, and included key information for each figure.
Discussion
Please provide a deeper discussion on the limitations and potential improvements in the methods.
Response the Reviewer comment: In the discussion, we have referenced the previous study by Yamamoto et al. (1998) [7], where they used non-dried callus for extraction, and mentioned that the high moisture content of the callus might lead to discrepancies with actual values. Additionally, in the conclusion, we highlighted the need for breeding using techniques such as gene editing.
We also acknowledged that the limited number of repetitions in this study led to high experimental variability, and we discussed the need for further experiments to perform more accurate research.
Conclusions
Conclusions in its current form rather resembles a summary of results. Please shorten the description of the results, limiting yourself to the most important conclusions of the paper. Please add information on further research perspectives.
Response the Reviewer comment: The conclusion has been rewritten to concisely present the key findings and include suggestions for future research directions.
This study demonstrates that methyl jasmonate (MeJA) is an effective elicitor for en-hancing verbascoside production in A. distichum callus cultures. While salicylic acid (SA) had no significant effect, treatment with 50 μM MeJA significantly increased verbascoside content, despite reductions in biomass. In bioreactor cultures, applying MeJA for 1 or 3 weeks resulted in the highest verbascoside yields, indicating that a short-term elicitation strategy may optimize both biomass and metabolite production. Notably, this study also confirms that the verbascoside content in callus cultures is comparable to that reported in leaves, suggesting the potential feasibility of using callus as an alternative source for verbascoside production. These findings provide a basis for developing an efficient in vitro production system for verbascoside. Further studies will be required to optimize detailed processes for potential commercial application, and it will also be necessary to develop superior varieties through gene editing.

Reviewer 3 Report
Comments and Suggestions for Authors
The manuscript is well written, and the experimental section is correctly executed. However, before the manuscript is accepted for publication some recommendations should be addressed:
Line 2. I propose the following title: “Effects of Elicitation on Abeliophyllum distichum Leaf Calli and Changes in Verbascoside Content”. The correct form is “content”.
Line 12: Specify what type of callus culture was used, whether in flasks, bioreactors, etc., because the comparison with the bioreactor system is not clear (Line 18).
Lines 41–51 show the advantages of in vitro cultures. Nevertheless, the manuscript does not address any disadvantage related to the culture system used for callus growth. It would be interesting to discuss the limiting factors that could affect verbascoside production.
Lines 63-68: Please clarify which in vitro culture systems were used, since Petri Dish culture was not mentioned in the abstract nor in the introduction.
Line 86: In Figure 2, please explain why the standard deviations are so high in the "Fresh weight" treatment.
Figures 2 and 3: Please revise the figures, with particular attention to enhancing the font size and refining the legend symbols that denote the various treatments.
Line 98: In Figure 3, please explain why the standard deviations are so high in the "MeJA" treatments.
Line 121: In Figure 5, Why was no statistical analysis applied to the results found for fresh weight?
Line 132: Figure 6. Most of the figures show data with high variability, particularly Figure 6. Could the authors expalin the reasons for this elevated deviation?
Line 141: Specify the verbascoside values reported in reference [7]. This comparison is important to understand the differences with previous work.
I suggest performing a correlation analysis between biomass (dry weight callus) and the verbascoside production (lines 121 and 131). Or the regression coefficient and correlation (R-square value, denoted by R2) from the correlation relationship between the two measurements (50μM MeJA).
Line 191: remove the period after "cm".
Line 192: remove the space between the numbers 2,4-D. This is the standard IUPAC-based name. Check this throughout the text.
Line 245: I recommend that the discussion section should be thoroughly revised, because as it is presented, it sounds like a review ot the results. Please make it clearer and more concise.
Author Response
<Reviewer 3>
The manuscript is well written, and the experimental section is correctly executed. However, before the manuscript is accepted for publication some recommendations should be addressed:
Line 2.I propose the following title: “Effects of Elicitation on Abeliophyllum distichum Leaf Calli and Changes in Verbascoside Content”. The correct form is “content”.
Response the Reviewer comment: Thanks for your comment. The title has been corrected from "contents" to "content."
Line 12: Specify what type of callus culture was used, whether in flasks, bioreactors, etc., because the comparison with the bioreactor system is not clear (Line 18).
Response the Reviewer comment: Thanks for your comment. It has been specified that the callus was cultured in Petri dishes (Line 14).
Lines 41–51 show the advantages of in vitro cultures. Nevertheless, the manuscript does not address any disadvantage related to the culture system used for callus growth. It would be interesting to discuss the limiting factors that could affect verbascoside production.
Response the Reviewer comment: Thanks for your comment. We have mentioned the significant cost involved in setting up the cultivation system (Line 49).
Lines 63-68: Please clarify which in vitro culture systems were used, since Petri Dish culture was not mentioned in the abstract nor in the introduction.
Response the Reviewer comment: It has been stated that Petri dish culture was used, and this is also mentioned in the abstract.
Line 86: In Figure 2, please explain why the standard deviations are so high in the "Fresh weight" treatment.
Response the Reviewer comment: Thanks for your comment. According to the report by Müller et al. (1990), callus cultures tend to exhibit higher instability compared to plant cultures, which can result in greater variability. Therefore, it is thought that the standard deviations were high in the "Fresh weight" treatment.
Figures 2 and 3: Please revise the figures, with particular attention to enhancing the font size and refining the legend symbols that denote the various treatments.
Response the Reviewer comment: The font size and legend symbol size for Figures 2 and 3 have been adjusted.
Line 98: In Figure 3, please explain why the standard deviations are so high in the "MeJA" treatments.
Line 121: In Figure 5, Why was no statistical analysis applied to the results found for fresh weight?Line 132: Figure 6. Most of the figures show data with high variability, particularly Figure 6. Could the authors expalin the reasons for this elevated deviation?
Response the Reviewer comment: Thanks for your comment. We have mentioned that the limited number of repetitions could lead to greater experimental variability. (Line 132)
Line 141: Specify the verbascoside values reported in reference [7]. This comparison is important to understand the differences with previous work.
Response the Reviewer comment: Thanks for your comment. The value of 7.15 mg/g of fresh callus from the previous study has been added (Line 157).
I suggest performing a correlation analysis between biomass (dry weight callus) and the verbascoside production (lines 121 and 131). Or the regression coefficient and correlation (R-square value, denoted by R2) from the correlation relationship between the two measurements (50μM MeJA).
Response the Reviewer comment: Thanks for your comment. As only three replicates were used in this study, performing a correlation analysis was difficult. In future research focused on commercialization, a correlation analysis will be conducted.
Line 191: remove the period after "cm".
Response the Reviewer comment: The period has been removed.
Line 192: remove the space between the numbers 2,4-D. This is the standard IUPAC-based name. Check this throughout the text.
Response the Reviewer comment: The extra space has been removed.
Line 245: I recommend that the discussion section should be thoroughly revised, because as it is presented, it sounds like a review ot the results. Please make it clearer and more concise.
Response the Reviewer comment: Thanks for your comment. The conclusion has been rewritten to present the key findings concisely, and suggestions for future research directions have been included.

Reviewer 4 Report
Comments and Suggestions for Authors
The authors of the manuscript ‘Effects of Elicitation on Abeliophyllum distichum Leaf Calli and Changes in Verbascoside Contents’ reported the impacts of salicylic acid and methyl jasmonate on callus growth and verbascoside content in the leaf callus cultures of Abeliophyllum distichum. They also used bioreactor cultures to compare the duration of methyl jasmonate treatment for the maximal production of verbascoside.
Specific comments:
L23: Please include the following keywords: methyl jasmonate; salicylic acid
L32: ‘numerous compounds’ Please indicate the compounds for better understanding.
L35: Please indicate the range of verbascoside content.
L58: Why did the authors choose salicylic acid and methyl jasmonate for this study?
L62: Please outline the drawbacks of the previous study [7]. Yamamoto, H.; Yoshida, K.; Kondo, Y.; Inoue, K. Production of cornoside in Abeliophyllum distichum cell suspension cultures. Phytochem. 1998, 48, 273-277.
L74: ‘calli showed browning’ is due to phenolic production or cell death; please specify.
L122: Please enhance the quality of Figure 5, and a mean separation is required for Figure 5a.
L127: Could you please explain the reasons behind the observed reduction in verbascoside content within the 2-week-old callus culture?
L186: Please indicate the age or growth stage of the plants.
L192-193: Citation required. It is known that media components, such as plant growth regulators, nutrient strength, and sucrose levels, affect the induction and proliferation of callus.
L207-208: Please indicate the type of bioreactor and the medium volume.
L238: Please specify the retention time and provide the supporting HPLC chromatograms for all samples to verify the changes in metabolites content.
L248-251: Please rewrite the sentence. SA or MeJA instead of SA ot MeJA
Comments on the Quality of English LanguageMinor corrections to the text are needed.
Author Response
<Reviewer 4>
The authors of the manuscript ‘Effects of Elicitation on Abeliophyllum distichum Leaf Calli and Changes in Verbascoside Contents’ reported the impacts of salicylic acid and methyl jasmonate on callus growth and verbascoside content in the leaf callus cultures of Abeliophyllum distichum. They also used bioreactor cultures to compare the duration of methyl jasmonate treatment for the maximal production of verbascoside.
Specific comments:
L23: Please include the following keywords: methyl jasmonate; salicylic acid
Response the Reviewer comment: Thanks for your comment. We added methyl jasmonate and salicylic acid to the keywords.
L32: ‘numerous compounds’ Please indicate the compounds for better understanding.
Response the Reviewer comment: Thanks for your comment. The comments have been revised as follows:
Among the numerous compounds such as coumaric acid, catechin identified in A. distichum, verbascoside, a caffeoyl phenylethanoid glycoside, also referred to as acteoside, is considered the key commercially important compound [8].
L35: Please indicate the range of verbascoside content.
Response the Reviewer comment: Thanks for your comment. We have described the verbascoside content in representative plants where it has been reported, and compared it with the content in A. distichum to highlight its competitive potential for verbascoside production.
L58: Why did the authors choose salicylic acid and methyl jasmonate for this study?
Response the Reviewer comment: Thanks for your comment. A reference has been added indicating that recent studies have frequently used SA and MeJA to increase secondary metabolite production in callus cultures.
L62: Please outline the drawbacks of the previous study [7]. Yamamoto, H.; Yoshida, K.; Kondo, Y.; Inoue, K. Production of cornoside in Abeliophyllum distichum cell suspension cultures. Phytochem. 1998, 48, 273-277.
Response the Reviewer comment: Thanks for your comment. We have referenced Yamamoto et al. (1998) [7], where non-dried callus was used for extraction, and noted that the high moisture content in callus may lead to discrepancies in actual values.
L74: ‘calli showed browning’ is due to phenolic production or cell death; please specify.
Response the Reviewer comment: Thanks for your comment. It has been specified that the callus browned and died due to high moisture content.
L122: Please enhance the quality of Figure 5, and a mean separation is required for Figure 5a.
Response the Reviewer comment: Thanks for your comment. The font size and legend symbol size in Fig. 5 have been increased to improve visibility.
L127: Could you please explain the reasons behind the observed reduction in verbascoside content within the 2-week-old callus culture?
Response the Reviewer comment: Thanks for your comment. We have stated that further studies, such as metabolic pathway analysis and the identification of genes encoding related enzymes, are needed to understand the limited increase in verbascoside content observed in the 2-week MeJA treatment group.
L186: Please indicate the age or growth stage of the plants.
Response the Reviewer comment: Thanks for your comment. It has been specified that the explants were taken from 3-year-old trees.
L192-193: Citation required. It is known that media components, such as plant growth regulators, nutrient strength, and sucrose levels, affect the induction and proliferation of callus.
Response the Reviewer comment: Thanks for your comment. References have been added (Line 217).
L207-208: Please indicate the type of bioreactor and the medium volume.
Response the Reviewer comment: The volume of the liquid medium used has been indicated as 2L. The use of a 5L balloon-type air-lift bioreactor (Dongmyung Chemical, Korea) has been specified.
L238: Please specify the retention time and provide the supporting HPLC chromatograms for all samples to verify the changes in metabolites content.
Response the Reviewer comment: Retention time has been added (15.35min). The chromatogram has been added as a Supplementary File.
L248-251: Please rewrite the sentence. SA or MeJA instead of SA ot MeJA
Response the Reviewer comment: Thanks for your comment. Typos have been corrected.

Reviewer 5 Report
Comments and Suggestions for Authors
Dear Authors,
Thank you for the opportunity to review the manuscript titled "Effects of Elicitation on Abeliophyllum distichum Leaf Calli and Changes in Verbascoside Contents".
The paper is well-structured, and the topic addressed is relevant to the field. I appreciate the efforts made in carrying out the experiments.
Some remarks:
Keywords:
Some keywords already appear in the title. I recommend either removing these keywords or replacing them with a more specific or related term that adds value and improves discoverability.
Abstract
Line 9: Line 9: change Abelophyllum to Abeliophyllum
Materials and Methods
Section 4.3: Methyl Jasmonate Elicitation Period in Bioreactor:
Please specify the following details regarding the bioreactor used in your experiments:
- type of bioreactor: (e.g., stirred tank, wave bioreactor, etc.);
- bioreactor Capacity: (e.g., total volume it can hold);
- volume of the culture medium used per bioreactor.
These specifics will help to clarify the experimental setup and facilitate reproducibility.
Results
Figure 5: Please add the letters indicating significant differences for the callus fresh weight (a).
Author Response
<Reviewer 5>
Dear Authors,
Thank you for the opportunity to review the manuscript titled "Effects of Elicitation on Abeliophyllum distichum Leaf Calli and Changes in Verbascoside Contents".
The paper is well-structured, and the topic addressed is relevant to the field. I appreciate the efforts made in carrying out the experiments.
Some remarks:
Keywords:
Some keywords already appear in the title. I recommend either removing these keywords or replacing them with a more specific or related term that adds value and improves discoverability.
Response the Reviewer comment: Thanks for your comment. Revised the keywords (Methyl jasmonate, Salicylic acid).
Abstract
Line 9: Line 9: change Abelophyllum to Abeliophyllum
Response the Reviewer comment: Thanks for your comment. Corrected the typo from "Abelophyllum" to "Abeliophyllum."
Materials and Methods
Section 4.3: Methyl Jasmonate Elicitation Period in Bioreactor:
Please specify the following details regarding the bioreactor used in your experiments:
- type of bioreactor: (e.g., stirred tank, wave bioreactor, etc.);
- bioreactor Capacity: (e.g., total volume it can hold);
Response the Reviewer comment: Thanks for your comment. Specified that a 5L balloon-type air-lift bioreactor (Dongmyung Chemical, Korea) was used (Line 221).
- volume of the culture medium used per bioreactor.
These specifics will help to clarify the experimental setup and facilitate reproducibility.
Response the Reviewer comment: Thanks for your comment. The volume of the liquid medium used is 2L, as stated (Line 220).
Results
Figure 5: Please add the letters indicating significant differences for the callus fresh weight (a).
Response the Reviewer comment: Thanks for your comment. The fresh weight in Fig. 5 (a) was not statistically significant, so no significance notation was made (Line 121).
